# Work in Progress: Teaching a Pipetting Task to a Robot Using Natural Gestures with Haptic Feedback in Augmented Reality

Max Alexej Pötter
Chair of Industrial Design
Engineering
Centre for Tactile Internet with
Human-in-the-Loop (CeTI)
Technische Universität Dresden
Dresden, Germany

Fabian Wagner
Chair of Industrial Design
Engineering
Technische Universität Dresden
Dresden, Germany

Jan Lagast
Chair of Industrial Design
Engineering
Technische Universität Dresden
Dresden, Germany

## ABSTRACT

Collaborative robotics in laboratory automation can foster reproducibility, efficiency, and innovation. Augmented reality methods are a promising approach to fostering intuitive interaction with collaborative robotic systems, This paper presents a robot teaching system for teaching a pipetting task to a cobot suited to a laboratory environment. The system uses a head-mounted display and a smart glove device offering an intuitive teaching method leveraging natural gestures and haptic feedback. We outline the current state of our work, detailing the implementation of different abstraction levels for the pipetting interaction assigning liquids for dispensation from beakers to microplates. Additionally, we present planned steps to finalize the system and introduce a study design aimed at better understanding the effects of abstraction levels in visualization and gestures on the user experience.

## CCS CONCEPTS

• **Human-centered computing** → **Mixed / augmented reality**; **Gestural input**; **Haptic devices**.

## KEYWORDS

human-robot interaction, augmented reality, gesture input, haptic feedback

**ACM Reference Format:**
Max Alexej Pötter, Fabian Wagner, and Jan Lagast. 2024. Work in Progress: Teaching a Pipetting Task to a Robot Using Natural Gestures with Haptic Feedback in Augmented Reality. In *Proceedings of ACM Conference (Conference'17)*. ACM, New York, NY, USA, 3 pages. https://doi.org/10.1145/nnnnnnn.nnnnnnn

## 1 INTRODUCTION

In recent years, robotics research has been focused on human-robot collaboration, exploring partnerships between humans and robots to achieve shared objectives. Collaborative robots, or cobots, emerge as versatile tools for multifaceted applications.

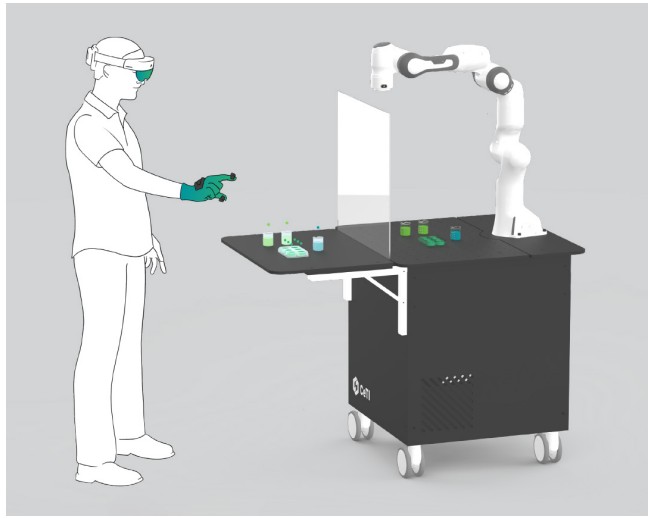

**Figure 1: Concept Visualization of the Robot Teaching System**

Laboratory environments are suited well for the integration of collaborative robotic systems. Here, the pursuit of automation aims not only to enhance efficiency but also to improve reproducibility [6], particularly in repetitive tasks and alleviating the burden of manual labor [2]. While specialized automation solutions for tasks like pipetting exist, the emergence of flexible cobot platforms for laboratory environments promises more flexible automation solutions, enabling novel research methodologies [6]. To interact with cobotic platforms, Augmented Reality (AR) emerges as an interaction modality facilitating intuitive communication and interaction [1, 5, 8] with robotic systems while also bridging the gap between physical and virtual environments seamlessly.

In alignment with these trends, our work focuses on the design and development of a robot teaching system for a pipetting task using a head-mounted display and a smart glove. The system enables the user to teach the robot using natural gestures and providing haptic feedback.

In the subsequent sections, we will present related work and the current state of our work. Furthermore, we will outline the next steps necessary to realize the system and propose a study design aimed at evaluating its User eXperience (UX).

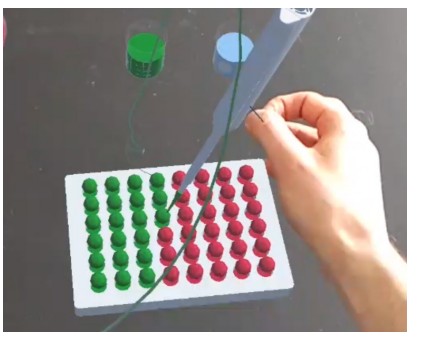
(a) Realistic Pipette

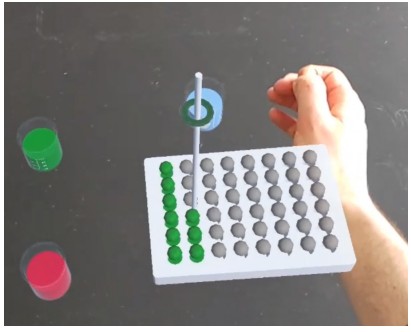
(b) Abstracted Pipette

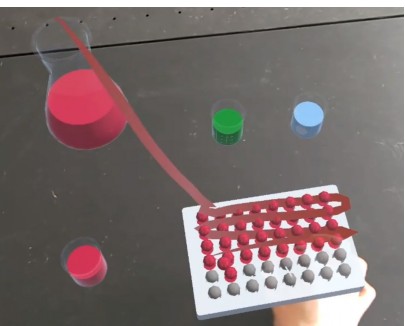
(c) Affordance Spheres

Figure 2: Pipetting interactions with varying levels of abstraction of the visualization

## 2 RELATED WORK

In recent years, researchers have proposed a multitude of AR systems for programming robots. In this paragraph, we will present an excerpt of the available literature. Chacko et al. [1] present a mobile AR framework to teach tool paths to a robot manipulator. The system was faster to use and caused less physical workload than teaching the robot by hand guidance. Fuste et al. [3] present a user interface framework for robot control in AR. Using AR allows the user to create and edit path programs while moving freely. Motion paths and speeds can be continuously visualized. Kapinus et al. [5] suggest a system to program processes to a robot by attaching virtual interaction cues to real objects and visualizing the process by a physical representation. Materna et al. [8] successfully employ spatial AR to provide a user interface tied to the workspace combining robot teaching on a process and path level for a stool assembly. The presented work shows the benefits of AR in Robot Teaching across a variety of use cases.

Regarding the application of robotic systems in laboratory environments, researchers successfully implemented commercial pipettes into end-effectors to automate pipetting using flexible and versatile robot arm systems. Knobbe et al. [6] demonstrated precise pipetting capabilities compliant with the ISO 8655 standard, leveraging a custom bracket to mount a pipette onto a Franka Emika Panda robot's standard gripper. Chaichaowarat et al. [2] proposed an economical robotic design integrating a commercial pipette into a SCARA robot platform, expanding access to automated pipetting solutions. Zhang et al. [12] implemented an end-effector with a manual pipette applying error correction using a visual system for low-frequency, high-repetition liquid dispensing tasks. These works showcase the feasibility of integrating manual pipettes in a robotic end effector of a collaborative robot and the benefits for application in laboratories.

## 3 THE ROBOT TEACHING SYSTEM

The system is going to be built up on a mobile cobot table structure. The robot used for the system will be a Universal Robots UR5e [10]. We originally planned to use a Franka Emika Panda, which is why it is depicted in fig.1. For the end effector of the robot, we will design a custom-made end effector integrating a pipette designed for manual use, similar to the design implemented by Zhang et al. [12]. The user will interact with the system by using a Microsoft HoloLens 2 [9]

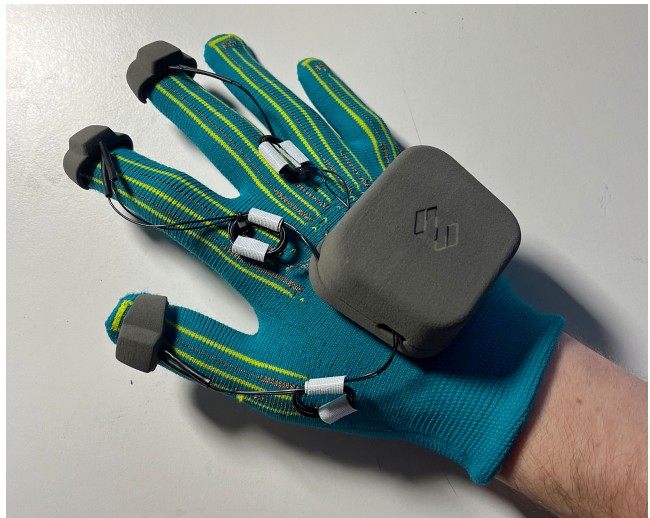

Figure 3: CeTI FingerTac Glove with textile integrated sensors for gesture control integrated with the FingerTac for haptic feedback

and a custom smart glove design (fig. 3). The user will be interacting in a virtual representation of the physical space of the robot, where digital twins of the physical labware are present. These can be manipulated to teach the robot the pipetting task. To enhance the system with additional gestures and to add haptic feedback, we will be using a smart glove device with textile-integrated strain sensors for finger bend measurements [11] and integrate it with the FingerTac, a haptic thimble device providing vibrotactile feedback specifically designed for AR applications [4]. Using the system, the user will be enabled to select liquids from beakers and assign the liquid to wells on a microplate. In the current state, we have designed variations of the pipetting interaction for the assignment of liquids.

## 4 PIPETTING INTERACTIONS

The proposed interaction patterns for the liquid assignment differ in the abstraction level of visualization (fig. 2b). Further, once we

integrate the CeTI FingerTac Glvoe with the robot teaching system, we will implement different gesture patterns for the interaction.

**Realistic Pipette.** The pipette is being visualized by a realistic pipette model (fig. 2a). To interact with the pipette, it will be grabbed using a fist-like gesture with the thumb elongated, resembling the gesture when interacting with a real pipette. To enable the pipette to pick up liquid, the thumb will be bent to press on the virtual plunger of the pipette accompanied by haptic feedback. Placing the pipette over a beaker and elongating the thumb, results in picking up the liquid from the beaker. After picking up liquid, the pipette can be dragged across the microplate to assign the liquid to wells in a rectangular pattern. To confirm the assignment, hence ´release the liquid´, the thumb is bent again to press on the plunger of the pipette, again augmented by haptic feedback.

**Abstracted Pipette.** The abstracted pipette is visualized as a bar with a torus on the upper end (fig. 2b). The gestures performed to pick up a liquid and assign it to wells of the microplate are the same as for the realistic pipette condition. The color fill of the torus represents the liquid that has been picked up.

**Affordance Spheres.** The affordance spheres condition (fig. 2c) further simplifies the interface and interaction pattern. Colored spheres float above the beakers. The spheres can be grabbed using a pinch gesture, which is provided by the Microsoft HoloLens 2 and used throughout many applications. The grabbed sphere can dragged across the microplate to assign the liquid to the wells again in a rectangular pattern.

## 5 OUTLOOK

The next steps in the development of the robotic system are the implementation of the robot, which we will do in a virtual environment for simulation purposes before we transition to setup the robot in the real envrionent. Forther, we will design an end effector integrating a manual pipette for executing the pipetting task. To enable the gesture patterns and the haptic feedback for the presented pipetting interactions, the CeTI FingerTac Glove will be integrated in the System.

To understand how the different abstraction levels of the pipetting interactions and the use of haptic feedback affect the UX, we will apply a 3x2 within-subject study design of the independent variables pipetting interaction (realistic pipette, abstracted pipette, affordance spheres) and haptic feedback (with, without). We suggest taking a mixed methods approach, collecting objective data on the duration and precision of performing the pipetting interaction, as well as subjective data provided by the participants. To evaluate the UX, we suggest applying the UX questionnaire (UEQ) [7] in the study design. To gather qualitatve insights on the interface, a semi-structured interview will be applied.

## ACKNOWLEDGMENTS

This work has been funded by the German Research Foundation (DFG, Deutsche Forschungsgemeinschaft) as part of Germany's Excellence Strategy – EXC 2050/1 – Project ID 390696704 – Cluster of Excellence "Centre for Tactile Internet with Human-in-the-Loop" (CeTI) of Technische Universität Dresden.

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
