# OpenReview forum: "Work in Progress: Teaching a Pipetting Task to a Robot Using Natural Gestures with Haptic Feedback in Augmented Reality"
_humanrobotinteraction.org/HRI/2024/Workshop/VAM-HRI — VAM-HRI 2024 Oral_

### Official Review · Reviewer_4zJ2 · 2024-02-22
**Accept**

**Rating:** 7
**Confidence:** 4

**Review:**

The paper "Work in Progress: Teaching a Pipetting Task to a Robot Using Natural Gestures with Haptic Feedback in Augmented Reality" by Max Alexej Pötter, Fabian Wagner, and Jan Lagast presents an innovative system for teaching pipetting tasks to collaborative robots (cobots) in laboratory settings using augmented reality (AR). This system utilizes a head-mounted display and a smart glove to provide intuitive interaction methods through natural gestures and haptic feedback. The authors outline the development and implementation of the system, emphasizing the importance of different abstraction levels for pipetting interaction and planning future studies to evaluate the effects of these levels on user experience.

## Strengths:
- **Innovative Approach:** Introduces a novel method for robot teaching in lab environments, emphasizing intuitive AR interactions.
- **Enhanced Interaction:** Utilizes natural gestures and haptic feedback, potentially improving task efficiency and learning curves.

## Weaknesses:
- **Work in Progress:** As the title suggests, the system is not yet fully implemented or evaluated, leaving uncertainties about its practical effectiveness.
- **Limited Focus:** The study's focus on pipetting tasks may limit its applicability to a broader range of robotic teaching applications.

## Recommendations for Improvement:
- **Broaden the Application Range:** Extend the system to cover more complex laboratory tasks beyond pipetting.
- **Comprehensive Evaluation:** Conduct thorough user studies to assess the system's effectiveness, user satisfaction, and learning outcomes in real-world settings.

In summary, I think this paper is a great fit for VAM-HRI, and I recommend acceptance.

---

### Official Review · Reviewer_RKbx · 2024-02-25
**Review B**

**Rating:** 8
**Confidence:** 5

**Review:**

Paper presents a mixed reality framework for robot learning from demonstration. The aim of this paper is taching the robot to operate autonomous laboratory.

Work, experiments and framework are well described. Authors presented a couple example tasks like pipetting. Both hardware and UI are well describied, designed and integrated.

One major weakness i see is the robot learning aspect itself. While everything is well documented, I do not see exactly how the robot is learning the task. Authors also do not discuss the algorithm that does it as well. RW also could use a bit more work and details

---

### Decision · Program_Chairs · 2024-02-26

Accept (Oral)